# Phenolic Composition and Antioxidant Properties of Cooked Rice Dyed with Sorghum-Leaf Bio-Colorants

**DOI:** 10.3390/foods10092058

**Published:** 2021-08-31

**Authors:** Franklin Brian Apea-Bah, Xiang Li, Trust Beta

**Affiliations:** 1Department of Food and Human Nutritional Sciences, Faculty of Agricultural and Food Sciences, University of Manitoba, Winnipeg, MB R3T 2N2, Canada; Franklin.Apea-Bah@umanitoba.ca; 2Department of Food Science, University of Guelph, Guelph, ON N1G 2W1, Canada; xli62@uoguelph.ca

**Keywords:** rice, cowpea, sorghum leaves, phenolic acids, flavonoids, anthocyanins, antioxidant properties

## Abstract

White rice is an important staple food globally. It is a rich source of energy but is low in dietary phenolic antioxidants. This current research aimed at providing scientific evidence for an alternative rice dish that has increased phenolic-antioxidant health-promoting potential by combining white rice with red cowpea beans and cooking with dye sorghum leaves hydrothermal extract, as a source of natural colorant. Boiled white rice and the rice–cowpea–sorghum extract dish were freeze-dried, and the free and bound phenolic compounds of raw and cooked samples were extracted. Phenolic composition, total phenolic content (TPC), and antioxidant activities (measured by 2,2-diphenyl-1-picrylhydrazyl radical scavenging capacity, Trolox equivalent antioxidant capacity, and oxygen radical absorbance capacity methods) of the raw and cooked samples were determined. Combining white rice with cowpea seeds and sorghum leaves extract significantly (*p* < 0.0001) increased the TPC and antioxidant activities of the rice due to the higher TPC and antioxidant activities of cowpea and sorghum leaves. Although boiling caused substantial losses of flavonoids and anthocyanins in the rice–cowpea–sorghum extract composite meal, the resulting dish had higher TPC and antioxidant activities than boiled white rice. Compositing white rice with phenolic-rich pulses can be an innovative approach to providing alternative healthy rice dishes to consumers.

## 1. Introduction

Rice (*Oryza sativa* L.) is the third most important cereal crop in the world after maize and wheat, with an estimated production of 501.1 million tons and an estimated total utilization of 502.5 million tons (milled equivalent) worldwide by 2019 [1]. It is a major staple food and principal dietary carbohydrate source for almost 50% of the world’s population [2]. Among the over 5000 varieties available in the world, differing in physical and chemical properties, Basmati rice appears to be the most popular and is marketed at high premium prices, especially in the USA and Europe. This is because the Basmati rice varieties have a desirable aroma, a pleasant taste, slender grains, length-wise elongation, and a tender cooked texture [2]. 

Although Basmati may come in various forms such as brown or white rice depending on the extent of milling, the white rice form is more common. This is because, in many Asian countries, brown rice was historically associated with poverty and food shortages and was mostly fed to sick people and the elderly. With increased consumer awareness about healthy eating, brown rice is now more expensive than white rice due to relatively low supply and challenges with storage and transportation [3].

During white rice production, the bran and all other outer layers are removed during the milling process. White rice is therefore rich in starch [4] but poor in other nutrients such as B-vitamins and essential minerals [3]. This has prompted several studies on fortification of white rice with vitamins and minerals and the technical considerations. It is also low in dietary phenolic compounds, which elicit antioxidant health benefits [5]. Some studies have reported a relationship between high consumption of white rice and chronic diseases. Hu et al. [4] observed, from reviewing several prospective cohort studies, an association between high white rice consumption and increased risk of type 2 diabetes. Epidemiological studies suggest a positive relationship between consumption of dietary phenolic compounds and reduction in the incidence of chronic diseases such as cardiovascular diseases [6], type 2 diabetes [7] and various forms of cancer [8]. In order to protect the consuming public from these debilitating chronic diseases, innovative approaches are needed to provide alternative healthy rice-based dishes to consumers.

Cereal–legume composite foods have been used for several years to combat malnutrition in vulnerable groups such as children in a number of countries. More recently, cereal–legume foods aimed at providing phenolic antioxidant benefits to adults are being explored [9]. Cowpea (*Vigna unguiculata*) is a pulse that is rich in protein [10] and flavonoids [11]. Compositing sorghum (*Sorghum bicolor*) with cowpea doubled the protein lysine content in the resulting prepared foods [12] and enhanced their phenolic antioxidant benefits [9]. Dye sorghum leaves are natural plant pigment sources with high antioxidant capacity due to their high anthocyanin content [13]. They have potential applications as food colorants [14]. 

The objective of this study was to provide scientific evidence for an alternative rice dish with increased antioxidant health-promoting potential by combining white rice with red cowpea seeds and cooking in dye sorghum leaves hydrothermal extract, as a source of natural colorant. 

## 2. Materials and Methods

### 2.1. Reagents

The phenolic standards comprising *p*-hydroxybenzaldehyde, protocatechuic aldehyde, 2,4-dihydroxybenzoic acid, *p*-coumaric acid, ferulic acid, catechin, quercetin and naringenin, as well as the chemicals 6-hydroxy-2,5,7,8-tetramethylchroman-2-carboxylic acid (Trolox), 2,2′-azino-bis(3-ethylbenzothiazoline-6-sulfonic acid) (ABTS), 2,2′-azobis(2-amidinopropane) dihydrochloride (AAPH), 2,2-diphenyl-1-picrylhydrazyl (DPPH), and fluorescein disodium salt, were acquired from Sigma-Aldrich Chemical Co. (St. Louis, MO, USA). Liquid chromatography-mass spectrometry (LCMS)-grade methanol, acetic acid, formic acid, LCMS-grade acetonitrile, Folin-Ciocalteu phenol reagent, hydrochloric acid, sodium hydroxide, dipotassium hydrogen phosphate, potassium dihydrogen phosphate, sodium carbonate, sodium iodide, ethyl acetate, and acetone were procured from Thermo Fisher Scientific (West Palm Beach, FL, USA). 

### 2.2. Source and Preparation of Samples

Three types of samples: Basmati white rice (*Oryza sativa*), red cowpea (*Vigna unguiculata*) seeds, and dye sorghum (Sorghum bicolor) leaves were procured from local shops in Winnipeg, Manitoba, Canada. A portion of the white rice was boiled in water to serve as the control sample. The test sample was prepared by first boiling dye sorghum leaves in water for 10 min, removing the leaves and boiling the cowpea seeds in the hydrothermal extract for 45 min. A portion of the raw white rice was then added, stirred to mix, and then boiled together to a soft texture. The rice, cowpea seeds, and dye sorghum leaves were used at the percent ratio of 134:90:1 *w*/*w*, respectively. The boiled samples were freeze-dried. All the samples comprising raw rice, raw cowpea seeds, raw dye sorghum leaves, the mixture of the three raw samples at a ratio of 134:90:1 *w*/*w* (rice: cowpea: dye sorghum leaves by weight), and the boiled samples were milled into flour to pass through a 500-µm sieve and stored at −20 °C. 

### 2.3. Extraction of Phenolic Compounds

The soluble free and insoluble bound phenolic acids in the milled samples were extracted using a modification of the method of Qiu, Liu, and Beta [15]. However, the flavonoids and anthocyanins were extracted using the method of Qiu, Liu, and Beta [16] with modification. For the phenolic acids, a 0.5 g weight of each sample (for dye sorghum leaves, 2.5 mg was used) was extracted twice with 6 mL of 2% (*v/v*) formic acid in methanol under sonication for 1 h. Each time, the suspension was centrifuged at 7000× *g* for 10 min at 10 °C and the supernatant collected. The pooled supernatant was evaporated to dryness at 35 °C and re-dissolved in 5 mL of 50% aqueous methanol. The residue was rinsed once with 6 mL of distilled, deionized water and saponified with 12 mL of 4 M NaOH for 4 h. It was then acidified with 6 M HCl to a pH of 1.5–2.0. The bound phenolics were extracted first with 30 mL of ethyl acetate and subsequently two more times, each with 20 mL ethyl acetate. The pooled extract was evaporated to dryness and re-dissolved with 2 mL of 50% aqueous methanol. 

For the flavonoids and anthocyanins, 0.5 g of the raw and cooked grains and 2.5 mg of the sorghum leaves were extracted once with 6 mL of acetone/water/formic acid (70:29:1) under sonication for 1 h. They were centrifuged as before and the collected supernatant evaporated to dryness. It was re-dissolved in 5 mL of 50% aqueous methanol. 

### 2.4. Analysis of Constituent Phenolic Compounds 

A Waters Alliance 2695 high performance liquid chromatograph (HPLC) (Milford, MA, USA) equipped with a quaternary pump, a Waters 2996 photodiode array detector, and a Waters 717 Plus autosampler were used to identify and quantify the phenolic compounds in the samples. The HPLC was coupled to a Micromass Micro^TM^ quadrupole time-of-flight mass spectrometer (Q-TOFMS) (Waters, Milford, MA, USA). The high performance liquid chromatograph-tandem mass spectrometer (HPLC-MS/MS) run for the phenolic acids was based on the method of Xiang, Apea-Bah, Ndolo, Katundu, and Beta [17], while the run for flavonoids and anthocyanins was based on the method of Qiu, Liu, and Beta [16]. For the phenolic acids, the mobile phase comprised: mobile phase A, 0.1% (*v/v*) aqueous formic acid, and mobile phase B, 0.1% (*v*/*v*) formic acid in methanol. The phenolic acids were eluted at a flow rate of 0.4 mL/min using the following gradient elution over a 25 min run: 0–3.81 min, 9–14% B; 3.81–4.85 min, 14–15% B; 4.85–5.89 min, 15% B; 5.89–8.32 min, 15–17% B; 8.32–9.71 min, 17–19% B; 9.71–10.40 min, 19% B; 10.40–12.48 min, 19–26% B; 12.48–13.17 min, 26–28% B; 13.17–14.21 min, 28–35% B; 14.21–15.95 min, 35–40% B; 15.95–16.64 min, 40–48% B; 16.64–18.37 min, 48–53% B; 18.37–22.53 min, 53–70% B; 22.53–22.88 min, 70–9% B; 22.88–25.00 min, 9% B. For the flavonoids and anthocyanins, however, a flow rate of 0.5 mL/min was used, and the gradient elution over 25 min was as follows: 0–2.5 min, 5–10% B; 2.5–7.5 min, 10–15% B; 7.5–10 min, 15–20% B; 10–15 min, 20–25% B; 15–20 min, 25–40% B; 20–22.5 min, 40–10% B; 22.5–25 min, 10–5% B. For both runs, an extract of 10 μL volume was injected onto an Accucore aQ, 100 × 3 mm, 2.6 μm column (Thermo Scientific, West Palm Beach, FL, USA). The column temperature was set at 35 °C, and the sample temperature was set at 15 °C. The Q-TOFMS was calibrated for the negative ion mode using NaI over the mass range of 100–1000 amu, with a resolution of 5000. Full mass spectra were recorded by using a capillary voltage of 1.45 kV and a cone voltage of 30 V. The flow rates of the cone gas (He_2_) and the desolvation gas (N_2_) were 45 L/h and 900 L/h, respectively. The desolvation gas and ion source temperatures were set at 250 °C and 120 °C, respectively. The collision energies were set at 15, 30, and 45 V to acquire the MS/MS spectra. The compounds were identified by comparing their retention times and ultraviolet (UV) and mass spectral characteristics with those of corresponding authentic phenolic standards, as well as compounds reported in the literature, where standards were unavailable. The identified compounds were quantified by comparing their peak areas with the authentic standards used to plot calibration curves at wavelengths of 280 nm for phenolic acids and phenolic aldehydes, and 350 nm for hydroxycinnamic acid derivatives, flavonoids, and 3-deoxyanthocyanidins. Protocatechuic aldehyde was quantified and expressed as protocatechuic acid equivalent; *p*-hydroxybenzaldehyde was quantified and expressed as *p*-hydroxybenzoic acid equivalent; coumaroylaldaric acids were quantified and expressed as *p*-coumaric acid equivalent; feruloyl aldaric acids, feruloyl methylaldaric acids, and 1,3-coumaroyl-feruloyl-glycerol were quantified and expressed as ferulic acid equivalent. Naringenin, eriodictyol, apigenin, luteolin, apigeninidin, and luteolinidin were quantified and expressed as naringin equivalent. Quercetin hexoside and quercetin dihexoside were quantified and expressed as quercetin equivalent. Catechin *O*-glucoside was quantified as (+)-catechin equivalent. The results were expressed as microgram per gram (µg/g) of sample, on a dry weight basis. MassLynx v. 4.1 software (Waters, Milford, MA, USA) was used for data acquisition.

### 2.5. Total Phenolic Content

Determination of the total phenolic content (TPC) of the soluble free and insoluble bound phenolic extracts was performed according to the method of Apea-Bah, Head, Scales, Bazylo, and Beta [18]. Ferulic acid was used to plot the standard curve, and results were presented as milligram ferulic acid equivalent per gram (mg FAE/g) milled sample on a dry weight basis.

### 2.6. Antioxidant Activities

Antioxidant activities of the soluble free and insoluble bound phenolic extracts were determined by measuring their radical scavenging capacities using the DPPH radical scavenging capacity, Trolox equivalent antioxidant capacity (TEAC), and oxygen radical absorbance capacity (ORAC) methods. DPPH radical scavenging capacity was measured using the method of Apea-Bah et al. [9] with absorbance read at 515 nm, while TEAC was measured using the method of Apea-Bah et al. [11]. ORAC was measured by the method of Qiu et al. [15] with a modification. The modification involved adding 150 µL of 0.816 nM fluorescein and 25 µL of 153 mM AAPH to 25 µL of the extract, blank (75 mM potassium phosphate buffer, pH 7.4) or Trolox standard (6.25–50 µM), and measuring the fluorescence decay (area under curve) at 37 °C over 50 min. The results for each of the three assays were presented as micromole Trolox equivalents per gram (μmol TE/g) sample on a dry weight basis.

### 2.7. Statistical Analysis

Results from all the analyses were presented as means ± standard deviations of triplicates. Two-way analysis of variance (ANOVA) was used to analyze the effects of sample type and replication on the response variables. Fisher’s least significant difference test was used to compare means that significantly (*p* < 0.0001) differed from each other. Principal component analysis was used to evaluate the association between the response variables. All statistical analyses were conducted with Statistica 10 (StatSoft Inc., Tulsa, OK, USA).

## 3. Results and Discussion

### 3.1. Phenolic Composition

Figure 1 is a chromatogram for raw rice, whilst Figure 2 is a chromatogram for raw cowpea beans. Figure 3 is a chromatogram for raw rice–cowpea–sorghum-leaves mix, whilst Figure 4 is a chromatogram for cooked rice–cowpea blend in sorghum leaves extract. Figure 5 is a chromatogram for raw sorghum leaves. Twenty five phenolic compounds were identified comprising hydroxybenzaldehydes (protocatechuic aldehyde and *p*-hydroxybenzaldehyde), hydroxybenzoic acid (2,4-dihydroxybenzoic acid), hydroxycinnamic acids and their derivatives (*p*-coumaric acid, ferulic acid, *p*-coumaroyl aldaric acids, feruloyl aldaric acids, and feruloyl methylaldaric acids), flavonoids (catechin *O*-glucoside, procyanidin B, quercetin *O*-dihexoside, quercetin *O*-hexoside, eriodictyol, naringenin, apigenin, and luteolin), and 3-deoxyanthocyanidins (luteolinidin and apigeninidin) (Table 1). These compounds were identified based on a comparison of their retention times and their UV and mass spectral data with those of authentic standards and published reports. For example, catechin *O*-glucoside, *p*-coumaroyl aldaric acids, feruloyl aldaric acids, feruloyl methylaldaric acids, procyanidin B dimer, (epi)afzelechin *O*-glucoside, quercetin *O*-dihexoside, and quercetin *O*-hexoside were identified by comparing their UV and mass spectral characteristics with those reported in cowpea by other researchers [9,11,19,20,21,22]. 

*p*-coumaroyl aldaric acid had a precursor ion [M-H]^−^ of *m/z* 355, which fragmented to release the product ions 209 (aldaric acid), 191 (209-18) (indicating a loss of H_2_O from the aldaric acid), and 147 (209-18-44), which indicated a loss of H_2_O and CO_2_ from the aldaric acid moiety. The wavelength for maximum absorption (λmax = 319) and mass spectral data were consistent with compounds reported by other researchers for cowpeas [20,21,22]. Feruloyl aldaric acid had a precursor ion of *m/z* 385, which fragmented to release the product ions, 209 (aldaric acid), 191 (209-18) (indicating a loss of H_2_O from the aldaric acid), and 147 (209-18-44), which indicated a loss of H_2_O and CO_2_ from the aldaric acid moiety. The wavelength for maximum absorption (λmax = 326 nm) and mass spectral data were consistent with compounds reported in the literature for cowpeas [20,21,22]. The λmax was also consistent with that of ferulic acid (322 nm) identified in the current study (Table 1). Feruloyl methylaldaric acid had a precursor ion of *m/z* 399, which fragmented to release the product ions 205, indicating a loss of ferulic acid (molecular weight 194); 191 (209-18), indicating a loss of H_2_O from aldaric acid (*m/z* 209); and 147 (209-18-44), which indicated a loss of H_2_O and CO_2_ from the aldaric acid moiety. The MS/MS fragments were consistent with similar compounds reported by Nderitu, Dykes, Awika, Minnaar, and Duodu [21]. A number of isomers have been reported in the literature, each for *p*-coumaroyl aldaric acid, feruloyl aldaric acid, and feruloyl methylaldaric acid in cowpeas [9,20,21,22]. As observed in the current study, the isomers for each compound were eluted at different retention times, and this is one major method to distinguish between the isomers. The isomers may differ from each other based on the point of attachment of the feruloyl moiety to the chiral centers of the aldaric acid. The hydroxycinnamic acids and their derivatives identified in cowpeas have been reported mostly as trans isomers [22]. 

The precursor ion [M-H]^−^ of catechin *O*-glucoside, *m/z* 451, fragmented to give a major product ion, *m/z* 289, indicative of catechin, with a loss of glucosyl moiety (451-162) [9,11,19,20,21,22]. The glycosylation position could not be determined. While Ojwang, Yang, Dykes, and Awika [19] indicated position 7, Hachibamba, Dykes, Awika, Minnaar, and Duodu [20] reported position 3 as the glycosylation position. Procyanidin B dimer had the precursor ion [M-H]^−^ of *m/z* 577 fragmenting to release the product ion 289, which is indicative of catechin, as explained earlier. (Epi)afzelechin *O*-glucoside had the precursor ion [M-H]^−^ of *m/z* 435 fragmenting to release a number of product ions including 299, 273, 161, 137, and 135, which are consistent with the findings of Ojwang et al. [19]. Ojwang et al. [19] identified two isomers of this compound, one of which had glycosylation position-3 on the flavonoid C-ring, and the other position-4′ on the B-ring. Although the compound identified in the current study matches more closely to the isomer with 3-*O*-glucoside based on the UV spectral characteristics, this assignment could not be confirmed in the current study. Therefore, there were three flavan-3-ols (catechin *O*-glucoside, procyanidin B dimer, and (epi)afzelechin *O*-glucoside) that were identified in the cowpea samples.

Quercetin dihexoside had a precursor ion of *m/z* 625 fragmenting to release the product ion 301, which is indicative of quercetin, with a loss of two hexosyl moieties (625-162-162). Similarly, quercetin hexoside had a precursor ion [M-H]^−^ of *m/z* 463, which fragmented to release the product ion 301, which is indicative of quercetin, with a loss of a hexosyl moiety (625-162-162). The major quercetin dihexosides reported in cowpeas are quercetin 3-*O*-galactosyl glucoside, quercetin 3-*O*-digalactoside, quercetin 3-*O*-diglucoside, and quercetin 3,7-diglucoside. Similarly, quercetin 3-*O*-galactoside, quercetin 7-*O*-glucoside, and quercetin 3-*O*-glucoside have been reported in cowpeas [19,20,21]. The exact hexosyl unit and glycosylation positions could not be determined in this study. The major flavonols identified in cowpeas were therefore quercetin dihexoside and quercetin hexoside. 

Naringenin had a precursor ion of *m/z* 271, which fragmented into product ions 177 and 151. Similarly, eriodictyol with precursor ion *m/z* 287 fragmented into product ions 193, 139, and 135. These were the flavanones identified in the sorghum leaves, and their products, and they have been previously reported in sorghum grains and sorghum–cowpea composite porridge [9]. The fragmentation pattern of the precursor ions was consistent with the reports of Apea-Bah [9].

Luteolin and apigenin were the flavones identified in the sorghum leaves and their products. Luteolin, with precursor ion *m/z* 285, fragmented into the product ion 133, while apigenin, with precursor ion *m/z* 269, fragmented into the product ions 117 and 149. These compounds with similar mass spectral data have been reported in sorghum grains, sorghum porridge, and sorghum–cowpea composite porridge [9]. Luteolinidin and apigeninidin were the 3-deoxyanthocyanidins that were identified in the sorghum leaves and its products. Luteolinidin, with a precursor ion *m/z* 269 did not show distinct fragmentation at the collision energy applied. Similarly, apigeninidin, with a precursor ion *m/z* 253, fragmented into the product ion 117. 

Table 2 shows the quantified results of the soluble free and insoluble bound phenolic acids and their derivatives in the samples. Raw white rice contained *p*-coumaric acid as the only free phenolic acid and both *p*-coumaric and ferulic acids as bound phenolic acids. Raw white rice contained 593% (7-fold) more ferulic acid than *p*-coumaric acid in the bound form. Raw cowpea had several isomers of *p*-coumaroyl aldaric acid, feruloyl aldaric acid, feruloyl methylaldaric acid, and 1,3-coumaroyl feruloyl glycerol in the soluble free fraction. It also contained protocatechuic aldehyde, 2,4-dihydroxybenzoic, *p*-coumaric, and ferulic acids in the bound form, with *p*-coumaric acid being the most abundant. Similar to the raw cowpea, the raw dye sorghum leaves had no free phenolic acids or their aldehydes but contained bound *p*-coumaric acid. When the raw samples were mixed together, the soluble free fraction contained *p*-coumaric and ferulic acids, while the bound phenolic aldehyde and acids were protocatechuic aldehyde, 2,4-dihydroxybenzoic, *p*-coumaric, and ferulic acids. 

The boiled white rice contained *p*-hydroxybenzaldehyde and *p*-coumaric acid as the soluble free phenolic acids as well as protocatechuic aldehyde, 2,4-dihydroxybenzoic, *p*-coumaric, and ferulic acids as the insoluble bound phenolic acids. The boiled rice–cowpea blend in sorghum extract contained only *p*-coumaric acid in the soluble free fraction and only ferulic acid in the insoluble bound fraction. Boiling the white rice therefore released *p*-hydroxybenzaldehyde in the soluble free fraction, as well as protocatechuic aldehyde and 2,4-dihydroxybenzoic acid in the insoluble bound form, in addition to the *p*-coumaric and ferulic acids initially present in the raw white rice. This may be due to hydrothermal-induced disruption of the rice grain matrix, which promoted the release of phenolic acids that may have been associated with other components in the grain.

In the rice–cowpea–sorghum leaves blend, however, boiling appeared to cause a loss of the ferulic acid present in the soluble free fraction of the raw mixed sample as well as the protocatechuic aldehyde, 2,4-dihydroxybenzoic acid, and *p*-coumaric acid initially present in the insoluble bound fraction of the raw mix. This may be due to phenolic–macronutrient interactions that might have occurred between the phenolic acids and macronutrients, such as proteins and lipids contributed by the cowpea, to the resulting cooked rice–cowpea blend. Phenolic–protein interactions have been studied extensively and are well-documented for phenolic acids and proteins [23,24], as well as flavonoids and proteins [25,26]. Fewer studies, however, have also reported phenolic interactions with other macronutrients such as carbohydrates and lipids, as reviewed by Bordenave, Hamaker, and Ferruzzi [27]. Since cowpeas are a good source of protein [10], it is presumable that cowpea proteins may have interacted covalently or non-covalently with some of the phenolic acids, making them non-extractable. 

Overall, the raw white rice had a 27% higher total soluble free phenolic acid content but a 163% lower total insoluble bound phenolic acid content than the raw mix. On boiling, however, white rice had a 36% higher total soluble free phenolic acid content and a 54% higher total insoluble bound phenolic acids content than the rice–cowpea blend boiled in the sorghum leaves extract. This indicates the positive effect of boiling on white rice with regards to phenolic acid content and composition. The addition of red cowpea to white rice and boiling in dye sorghum leaves extract therefore negatively affected both the free and bound phenolic acid content and the composition of the resulting food. 

It is worth noting that, overall, raw white rice had a 428% (5-fold) lower total soluble free phenolic acid content than the total insoluble bound phenolic acid content. Moreover, the boiled white rice had a 412% (5-fold) lower total soluble free phenolic acid content than the total insoluble bound phenolic acid content. Similarly, the raw rice–cowpea–sorghum leaves mix had a 1816% (7-fold) lower total soluble free phenolic acid content than the total insoluble bound phenolic acid content. The cooked rice–cowpea blend had a 373% (4-fold) lower total soluble free phenolic acid content than the total insoluble bound phenolic acid content. This underscores the important contribution of bound phenolic acids to the overall phenolic acid content and the composition of grain-based foods. When food is consumed and digested, it is anticipated that the soluble free phenolic acids, if not degraded by the digestive system (enzymes and pH), will be absorbed in the small intestine into systemic circulation, while the insoluble bound phenolic acids will remain unchanged and be transported to the large intestine where they will be catabolized by the colonic microbiota through fermentation. Portions of the colonic metabolites will then be absorbed into systemic circulation, while the remaining will maintain the antioxidant health of the colon.

Table 3 shows the flavonoids and anthocyanins identified and quantified in the samples. Generally, all the flavonoids and anthocyanins were identified in the soluble free fraction with none identified in the insoluble bound form. No flavonoids and anthocyanins were identified in the raw and boiled white rice. The raw red cowpea contained catechin *O*-glucoside, procyanidin B, (epi)-afzelechin *O*-glucoside, quercetin *O*-dihexoside, and quercetin *O*-hexoside, with catechin *O*-glucoside being the dominant flavonoid in the cowpea. These compounds have been reported previously in several cowpea genotypes [9,28]. Ojwang et al. [19] reported catechin *O*-glucoside to be the dominant flavonoid in a proanthocyanidin-rich extract from cowpea. 

The raw dye sorghum leaves, on the other hand, contained the 3-deoxyanthocyanidins: luteolinidin and apigeninidin, as well as the flavanones: naringenin and eriodictyol and the flavones: luteolin and apigenin (Table 3). This is in agreement with Kayodé et al. [13] and Kayodé et al. [29] who reported luteolinidin and apigeninidin as the major 3-deoxyanthocyanidins in leaf sheaths of dye sorghum. The concentration of apigeninidin in the raw dye sorghum leaves was 21-fold higher than that of luteolinidin, and this is consistent with Kayodé et al. [29] who reported the mean apigeninidin content from leaf sheaths of six sorghum varieties to be 25-fold higher than the mean content of luteolinidin. While naringenin and eriodictyol have been reported as the major flavanones in sorghum grains [9,30,31], this appears to be their first report in sorghum leaves. Eriodictyol content in the dye sorghum leaves was 5-fold higher than that of naringenin. Apigenin and luteolin have also been previously reported in sorghum grains [9,30,31], but this appears to be their first report in sorghum leaves. The apigenin content in the sorghum leaves was 3-fold higher than that of luteolin (Table 3).

As expected, combining the raw white rice, red cowpea, and dye sorghum leaves therefore yielded an extract comprising all the flavonoids and anthocyanins from the cowpea and sorghum leaves (Table 3). The boiling of the rice–cowpea blend in the sorghum leaves extract caused a complete loss of naringenin and luteolinidin, a 3-fold loss in catechin the *O*-glucoside content, a 1.6-fold loss in the quercetin *O*-dihexoside content, a 2-fold loss in the eriodictyol content, and an 88-fold loss in the apigeninidin content of the resulting cooked rice–cowpea blend. As explained earlier, the losses in the polyphenols may probably be due to either thermal degradation or their interactions with proteins [26,32] contributed by cowpea to the cooked rice–cowpea blend. Such polyphenol–protein interactions make them non-extractable, especially for covalent interactions. While there is sparse information on the interactions between 3-deoxyanthocyanidins and proteins, such interactions are plausible due to the available OH-groups on the A-ring and B-ring as well as the points of unsaturation on the C-ring [26,27]. It is worth noting, however, that (epi)afzelechin *O*-glucoside content substantially increased by 73% on cooking the rice–cowpea mix in the sorghum leaves extract. This may be due to thermal degradation of the plant tissue, thereby enhancing release of the compound. Different phenolic compounds react differently during cooking. While the concentration of some compounds decrease during cooking, others increase [9]. Although substantial losses in flavonoids and 3-deoxyanthocyanins occurred upon cooking of the rice–cowpea blend in dye sorghum leaves extract, the resulting food retained some of the polyphenols that may contribute to its antioxidant properties. 

### 3.2. Total Phenolic CONTENT and Antioxidant Activity 

Table 4 shows the total phenolic content (TPC) and antioxidant activities of the samples. Antioxidant activities were measured using three assays: (i) the DPPH radical scavenging activity, (ii) the Trolox equivalent antioxidant capacity (TEAC), and (iii) the oxygen radical absorbance capacity (ORAC) methods. Both DPPH and TEAC assays measure the ability of an antioxidant to reduce a reactive free radical or oxidant by transferring an electron to the radical or oxidant. This is also called the electron transfer mechanism (ET) [33]. ORAC, on the other hand, measures the ability of an antioxidant to compete with a substrate for an oxidant, which may be thermally generated in the in vitro reaction mixture. Here, the antioxidant donates a hydrogen atom to bind with the reactive oxidant, thereby stabilizing or quenching it. This is the hydrogen atom transfer (HAT) mechanism [33]. The use of all three methods demonstrates the robustness and versatility of the antioxidant system in quenching oxidants.

In comparing the soluble free phenolics fraction of the raw samples, dye sorghum leaves had the highest TPC and antioxidant activities followed by red cowpea seeds and then white rice. This is consistent with the polyphenol composition of the samples (Table 3) which showed the dye sorghum leaves to have very high levels of the 3-deoxyanthocyanidins: luteolinidin and apigeninidin, as well as high levels of the flavanones: naringenin and eriodictyol, followed by cowpea, which contained catechin *O*-glucoside and quercetin *O*-dihexoside. White rice, on the other hand, contained no flavonoids. It is important to note that the flavonoids and 3-deoxyanthocyanidins present in the soluble free fraction of the dye sorghum leaves and cowpea compensated for their lack of free phenolic acids. The antioxidant properties of cowpea flavonoids [11] and sorghum leaves 3-deoxyanthocyanidins [13,29] have been well documented. When white rice was combined with the cowpea and dye sorghum leaves, the TPC of the resulting raw rice–cowpea–sorghum-leaves mix was 12-fold higher than the corresponding raw white rice. Similarly, the antioxidant activities of the raw rice–cowpea–sorghum-leaves mix was 17–41-fold higher than the corresponding raw white rice (Table 3). This observation is consistent with the report of Apea-Bah et al. [9], who demonstrated the comparative advantage of compositing cereals (sorghum and maize) with legumes (cowpea and soybean), with respect to increasing the antioxidant activity of the cereals.

The TPC and antioxidant activities of the insoluble bound phenolic fraction for the raw samples followed the same trend as that of the soluble free phenolic fraction. Dye sorghum leaves had a very high level of *p*-coumaric acid, which was its only phenolic acid identified (Table 2). Cowpea, on the other hand, contained a variety of hydroxybenzaldehyde, hydroxybenzoic acid, and hydroxycinnamic acids with the total content being more than 2-fold of the total insoluble bound phenolic acid content of the white rice (Table 2). Consequently, the addition of the cowpea and the dye sorghum leaves to the white rice increased its TPC 6-fold and its antioxidant activities 3–7-fold, by enhancing the phenolic composition of the resulting rice–cowpea–sorghum-leaves mix. 

Boiling generally had no effect on the TPC and antioxidant activities of the soluble free and insoluble bound phenolic fractions of white rice, with the exception of the ORAC value of the insoluble bound fraction that decreased. For the rice–cowpea–sorghum-leaves mixture, however, boiling generally decreased the TPC and antioxidant activities of the soluble free phenolic fraction, while the insoluble bound fraction showed an inconsistent trend. While the TPC and DPPH radical scavenging capacity of the insoluble phenolic fraction of the resulting boiled rice–cowpea blend remained unchanged upon boiling, the TEAC value increased while the ORAC value decreased. The inconsistency in the effect of cooking on the phenolic content and the antioxidant activities has also been reported by Taylor and Duodu [34]. 

It is important to note, however, that consistent with the phenolic composition of the samples, the TPC and antioxidant activities of the soluble free and insoluble bound phenolic fractions of the boiled rice–cowpea blend were higher than that of the boiled white rice. To put this into perspective, the TPC of the soluble free and insoluble bound phenolic fractions of the boiled rice–cowpea blend were 9-fold and 6-fold higher compared to the values recorded for boiled white rice. Similarly, the antioxidant activities of the soluble free phenolic fraction of the boiled rice–cowpea blend was 9–15-fold higher than that of the corresponding fraction of white rice, while the antioxidant activities of the insoluble bound phenolic fraction of the boiled rice–cowpea blend was 6–9-fold higher than that of the corresponding fraction of white rice.

### 3.3. Principal Component Analysis

Principal component analysis of the dependent variables, based on the correlation matrix, led to the extraction of five principal components with measurable Eigen values. Three of the principal components had Eigen values greater than 1, and, together, they accounted for 95% of the total variability in the dependent variables (Figure 6). On the horizontal axis of the factor loading and score plot (Figure 7), the first principal component, which accounted for 68% of the total variability, separated all the dependent variables into two groups. One of the groups comprised all the 3-deoxyanthocyanidins and flavonoids, except the quercetin glycosides, and correlated closely with the total phenolic content, antioxidant activities, and the total quantified flavonoids. The remaining variables on the opposite side of the horizontal axis, comprised mostly phenolic acids, their derivatives, and the quercetin glycosides. In terms of sample contribution to the principal components, raw sorghum leaves contributed 82.7% to principal component 1 (Table 5). It may be inferred that principal component 1 demonstrated the major contribution of the sorghum leaves’ phenolic compounds to the antioxidant activities. This is further demonstrated by the sorghum leaves being separated from all the other samples and lying on the same side of principal component 1, where its constituent phenolic compounds clustered (Figure 7 and Appendix A).

Principal component 2, which accounted for 19.8% of the total variability, separated the samples into two groups such that raw cowpea and the raw rice–cowpea–sorghum-leaves mixture clustered together, while the raw rice, the cooked rice, and the cooked rice–cowpea blend in sorghum extract also clustered together. This segregation may be related to the total amount of quantified phenolic compounds in the respective groups of samples. Raw cowpea and the raw samples mix had higher overall quantified phenolic compounds concentrations than the raw rice, the cooked rice, and the cooked rice–cowpea blend. Raw cowpea contributed 63.5% to principal component 2 (Table 5). The results from the principal component analysis underpin the unique advantage of incorporating sorghum leaves extract and cowpea into a rice dish, from an antioxidant perspective. 

## 4. Conclusions

This research demonstrated that *p*-coumaric acid is the only soluble free phenolic acid present in white rice, while *p*-coumaric acid and ferulic acid are the bound phenolic acids present in white rice. Boiling disrupts the plant tissues and promotes the release of more free and bound phenolic acids in white rice. Combining white rice with red cowpea beans and dye sorghum leaves increases the TPC and antioxidant activities of the resulting mixture due to flavonoids and 3-deoxyanthocyanins contributed by the cowpea seeds and dye sorghum leaves. Boiling decreases the TPC and antioxidant activities of the rice–cowpea–sorghum leaves’ extract composite blend, due to losses in the flavonoid and 3-deoxyanthocyanin contents. The resulting product, notwithstanding, retains some of the constituent polyphenols, which contribute to its phenolic content and antioxidant activities. Compositing white rice with phenolic-rich pulses and natural plant pigments can be an innovative approach to providing alternative healthy rice dishes for consumers.

## Figures and Tables

**Figure 1 foods-10-02058-f001:**
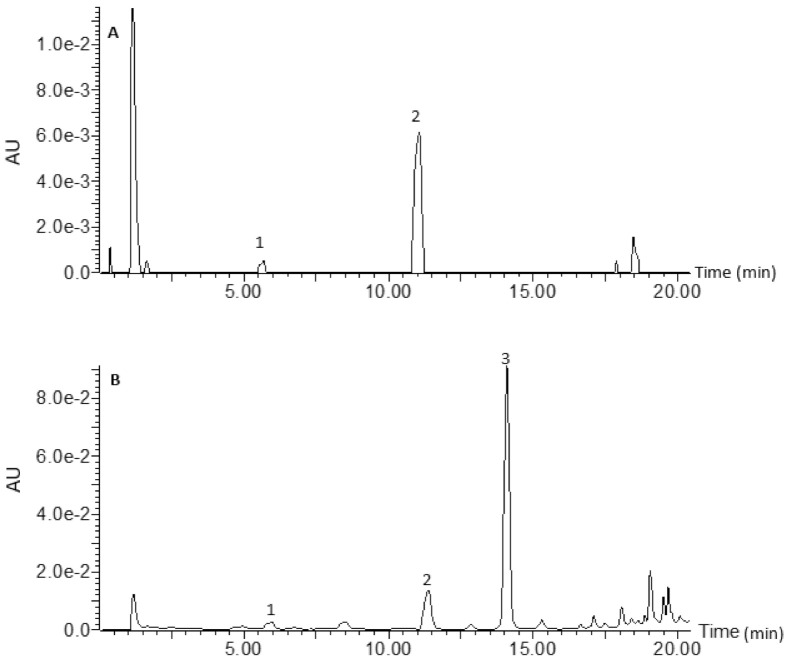
Liquid chromatogram for raw rice at 280 (**A**) nm and 350 nm (**B**). Key: 1 = *p*-hydroxybenzaldehyde; 2 = *p*-coumaric acid; 3 = ferulic acid.

**Figure 2 foods-10-02058-f002:**
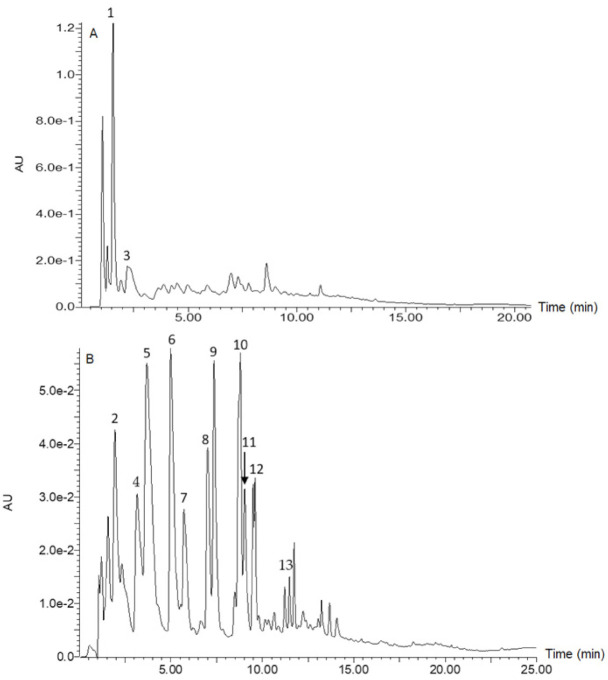
Liquid chromatogram for cowpea at 280 nm (**A**) and 350 nm (**B**). Key: 1 = catechin glucoside (RT = 1.55 min, *m/z* 451 amu); 2 = *p*-coumaroylaldaric acid (RT = 1.93 min, *m/z* 355 amu); 3 = procyanidin B1 (RT = 2.33 min, *m/z* 577 amu); 4 = *p*-coumaroylaldaric acid (RT = 3.25 min, *m/z* 355 amu); 5 = feruloylaldaric acid (RT = 3.70 min, *m/z* 385 amu); 6 = feruloylaldaric acid (RT = 5.10 min, *m/z* 385 amu); 7 = feruloyl methylaldaric acid (RT = 5.73 min, *m/z* 399 amu); 8 = feruloyl methylaldaric acid (RT = 7.04 min, *m/z* 399 amu); 9 = feruloyl methylaldaric acid (RT = 7.36 min, *m/z* 399 amu); 10 = (Epi)afzelechin *O*-glucoside (RT = 8.71 min, *m/z* 435 amu); 11 = quercetin dihexoside (RT = 9.48 min, *m/z* 625 amu); 12 = 1,3-coumaroyl-feruloyl-glycerol (RT = 9.58 min, *m/z* 413 amu); 13 = quercetin hexoside (RT = 11.75 min, *m/z* 463 amu).

**Figure 3 foods-10-02058-f003:**
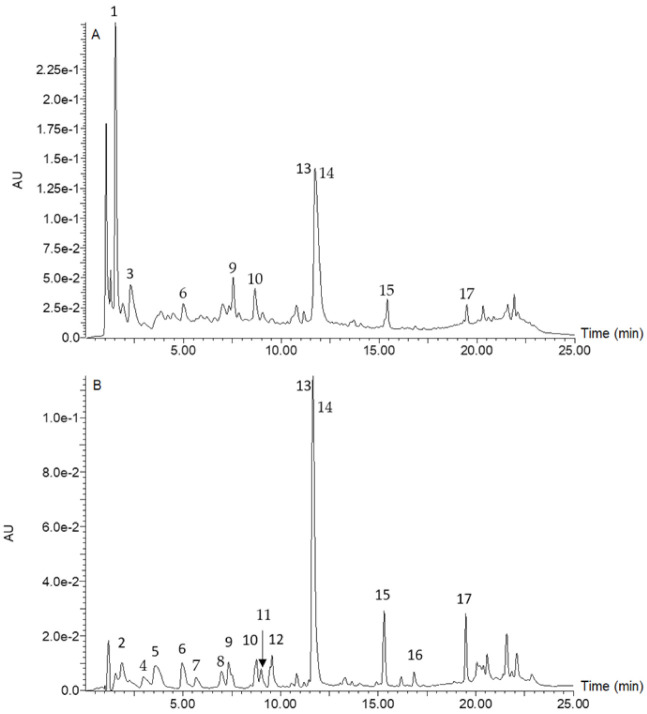
Liquid chromatogram for raw rice–cowpea–sorghum leaves mix at 280 nm (**A**) and 350 nm (**B**). Key: 1 = catechin glucoside (RT = 1.55 min, *m/z* 451 amu); 2 = *p*-coumaroylaldaric acid (RT = 1.93 min, *m/z* 355 amu); 3 = procyanidin B1 (RT = 2.33 min, *m/z* 577 amu); 4 = *p*-coumaroylaldaric acid (RT = 3.25 min, *m/z* 355 amu); 5 = feruloylaldaric acid (RT = 3.70 min, *m/z* 385 amu); 6 = feruloylaldaric acid (RT = 5.10 min, *m/z* 385 amu); 7 = feruloyl methylaldaric acid (RT = 5.85 min, *m/z* 399 amu); 8 = feruloyl methylaldaric acid (RT = 7.15 min, *m/z* 399 amu); 9 = feruloyl methylaldaric acid (RT = 7.35 min, *m/z* 399 amu); 10 = (Epi)afzelechin *O*-glucoside (RT = 8.71 min, *m/z* 435 amu); 11 = quercetin dihexoside (RT = 9.40 min, *m/z* 625 amu); 12 = 1,3-coumaroyl-feruloyl-glycerol (RT = 9.58 min, *m/z* 413 amu); 13 = apigeninidin (RT = 11.45 min, *m/z* 253 amu); 14 = naringenin (RT = 11.65 min, *m/z* 271 amu); 15 = eriodictyol (RT = 15.45 min, *m/z* 287 amu); 16 = luteolin (RT = 16.85 min, *m/z* 285 amu); 17 = apigenin (RT = 19.51 min, *m/z* 269 amu). Apigeninidin was coeluted with naringenin, and so naringenin was identified and quantified at 350 nm, while apigeninidin was identified and quantified at 450 nm.

**Figure 4 foods-10-02058-f004:**
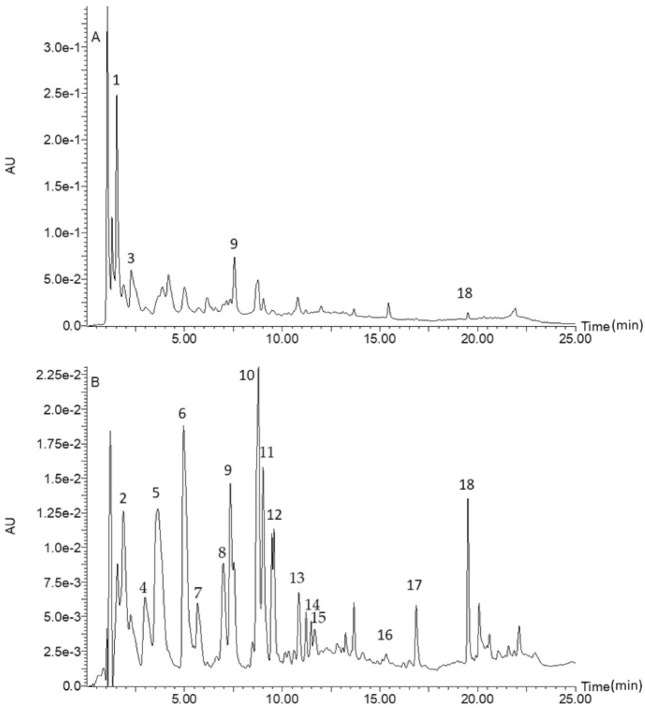
Liquid chromatogram for cooked rice–cowpea blend in sorghum leaves extract at 280 nm (**A**) and 350 nm (**B**). Key: 1 **=** catechin glucoside (RT = 1.55 min, *m/z* 451 amu); 2 = *p*-coumaroylaldaric acid (RT = 1.93 min, *m/z* 355 amu); 3 = procyanidin B (RT = 2.33 min, *m/z* 577 amu); 4 = *p*-coumaroylaldaric acid (RT = 3.25 min, *m/z* 355 amu); 5 = feruloylaldaric acid (RT = 3.70 min, *m/z* 385 amu); 6 = feruloylaldaric acid (RT = 5.10 min, *m/z* 385 amu); 7 = feruloyl methylaldaric acid (RT = 5.85 min, *m/z* 399 amu); 8 = feruloyl methylaldaric acid (RT = 7.15 min, *m/z* 399 amu); 9 = feruloyl methylaldaric acid (RT = 7.35 min, *m/z* 399 amu); 10 = (Epi)afzelechin *O*-glucoside (RT = 8.71 min, *m/z* 435 amu); 11 = quercetin dihexoside (RT = 9.48 min, *m/z* 625 amu); 12 = 1,3-coumaroyl-feruloyl-glycerol (RT = 9.58 min, *m/z* 413 amu); 13 = quercetin hexoside (RT = 11.28 min, *m/z* 463 amu); 14 = apigeninidin (RT = 11.65 min, *m/z* 253 amu); 15 = naringenin (RT = 11.75 min, *m/z* 271 amu); 16 = eriodictyol (RT = 15.45 min, *m/z* 287 amu); 17 = luteolin (RT = 16.85 min, *m/z* 285 amu); 18 = apigenin (RT = 19.51 min, *m/z* 269 amu). Apigeninidin coeluted with naringenin, and so naringenin was identified and quantified at 350 nm, while apigeninidin was identified and quantified at 450 nm.

**Figure 5 foods-10-02058-f005:**
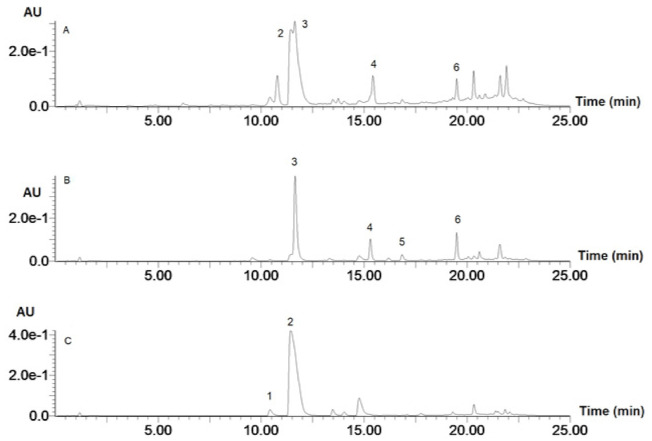
Liquid chromatogram for sorghum leaves at 280 nm (**A**), 350 nm (**B**), and 450 nm (**C**). Key: 1 **=** luteolinidin (RT = 10.42 min, *m/z* 269 amu); 2 = apigeninidin (RT = 11.43 min, *m/z* 253 amu); 3 = naringenin (RT = 11.63 min, *m/z* 271 amu); 4 = eriodictyol (RT = 15.43 min, *m/z* 287 amu); 5 = luteolin (RT = 16.85 min, *m/z* 285 amu); 6 = apigenin (RT = 19.48 min, *m/z* 269 amu). Apigeninidin was coeluted with naringenin, and so naringenin was identified and quantified at 350 nm, while apigeninidin was identified and quantified at 450 nm.

**Figure 6 foods-10-02058-f006:**
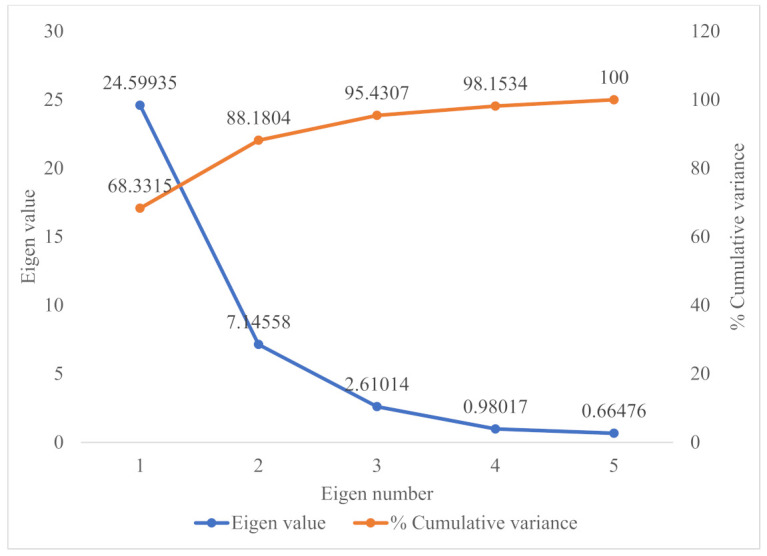
Scree plot of the principal components for raw white rice, cowpea, sorghum leaves, their composite blend, and cooked products, showing Eigen values and contribution to total variance.

**Figure 7 foods-10-02058-f007:**
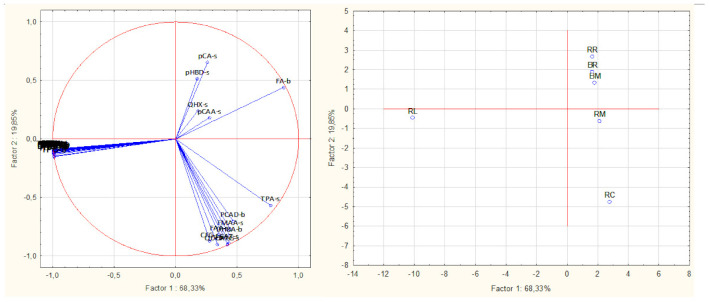
Factor loading and score plots for the phenolic compounds and antioxidant activities (**left**) and samples (**right**) during principal components analysis. Key: RR = raw white rice; RC = raw cowpea; RL = raw leaves; RM = raw mix; BR = boiled white rice; BM = boiled rice-cowpea mix; pCA-s = *p*-coumaric acid soluble free; pHBD-s = *p*-hydroxybenzaldehyde soluble free; QHX-s = quercetin hexoside soluble free; CAA = *p*-coumaroyl aldaric acid; TPA = quantified total phenolic acids; PCAD = protocatechuic aldehyde; FMAA = feruloyl methylaldaric acid; FAA = feruloyl aldaric acid; X-s = compound X soluble free fraction; pHBA = *p*-hydroxybenzoic acid; CFG = 1,3-Coumaroyl feruloyl glycerol; X-b = compound X insoluble bound fraction.

**Table 1 foods-10-02058-t001:** UV spectral characteristics, mass spectral data, and molecular formulae of phenolic compounds identified in raw white rice, cowpea, sorghum leaves, and their composite blend and cooked products.

No.	λmax, nm	[M-H]^−^	MS/MS	Molecular Formula	Compound
1	210, 279	451	289 (100), 451 (17)	C_21_H_24_O_11_	Catechin 3-*O*-glucoside
2	210, 280	137	137 (100), 101 (33), 109 (33)	C_7_H_6_O_3_	Protocatechuic aldehyde
3	210, 282	577	109 (100), 289 (68), 123 (63), 245 (44)	C_15_H_14_O_6_	Procyanidin B
4	284	121	121 (100)	C_7_H_6_O_2_	*p*-hydroxybenzaldehyde
5	210, 289	153	131 (100), 147 (28)	C_7_H_6_O_4_	2,4-dihydroxybenzoic acid
6	210, 308	163	119 (100)	C_9_H_8_O_3_	*p*-coumaric acid
7	217, 322	193	134 (100), 178 (30)	C_10_H_10_O_4_	Ferulic acid
8	210, 319	355	209 (100), 191 (45), 147 (21)	C_15_H_16_O_10_	*p*-coumaroyl aldaric acid
9	210, 319	355	209 (100), 191 (45), 147 (21)	C_15_H_16_O_10_	*p*-coumaroyl aldaric acid
10	210, 326	385	191 (100), 209 (57), 147 (21)	C_16_H_18_O_11_	Feruloylaldaric acid
11	210, 326	385	191 (100), 209 (54), 147 (24)	C_16_H_18_O_11_	Feruloylaldaric acid
12	210, 326	385	191 (100), 209 (61), 147 (22)	C_16_H_18_O_11_	Feruloylaldaric acid
13	210, 327	399	191 (100), 173 (46), 147 (27)	C_17_H_20_O_11_	Feruloyl methylaldaric acid
14	210, 327	399	191 (100), 173 (46), 147 (27)	C_17_H_20_O_11_	Feruloyl methylaldaric acid
15	210, 327	399	191 (100), 173 (46), 147 (27)	C_17_H_20_O_11_	Feruloyl methylaldaric acid
16	210, 320	413	191 (100), 173 (56), 147 (21)	C_22_H_22_O_8_	1,3-coumaroyl-feruloyl-glycerol
17	210, 279	435	435 (100), 137 (49), 299 (48), 161 (20), 135 (15), 273 (4)	C_21_H_24_O_10_	(Epi)afzelechin *O*-glucoside
18	211, 269, 370	625	625 (100), 300 (79), 301 (56)	C_27_H_30_O_17_	Quercetin *O*-dihexoside
19	212, 371	271	151 (100), 177 (30), 271 (12)	C_15_H_12_O_5_	Naringenin
20	210, 321	463	463 (100), 301 (83)	C_21_H_20_O_12_	Quercetin *O*-hexoside
21	212, 279, 487	269	269 (100)	C_15_H_11_O_5_^+^	Luteolinidin
22	210, 277	287	139 (100), 193 (39), 287 (17)	C_15_H_12_O_6_	Eriodictyol
23	210, 276, 320, 475	253	253 (100), 117 (30)	C_15_H_11_O_4_^+^	Apigeninidin
24	214, 335	285	285 (100), 133 (59)	C_15_H_10_O_6_	Luteolin
25	215, 267, 337	269	117 (100), 149 (27), 269 (13)	C_15_H_10_O_5_	Apigenin

Key: No. = arbitrary number for counting compounds identified; λmax = UV wavelength at maximum absorption; [M-H]^−^ = precursor ions; MS/MS = product ions. Hydroxybenzoic acids were identified at 280 nm; hydroxycinnamic acids, their derivatives, and flavonoids at 350 nm; and 3-deoxyanthocyanidins were identified at 450 nm.

**Table 2 foods-10-02058-t002:** Phenolic acid composition of raw white rice, cowpea, sorghum leaves, their composite blend, and cooked products.

Sample	Raw Rice	Raw Cowpea	Raw Leaves	Raw Mix	Boiled Rice	Cooked Rice–Cowpea Blend	Total
Soluble free phenolics							
*p*-hydroxybenzaldehyde	22.1 ± 0.2 ^b^	nd	nd	nd	1.5 ± 0.0 ^a^	nd	23.6 ± 0.2 ^D^
*p*-coumaric acid	20.4 ± 0.6 ^a^	nd	nd	nd	22.4 ± 2.3 ^a^	nd	42.8 ± 2.1 ^F^
*p*-coumaroyl aldaric acid	nd	13.8 ± 1.2 ^b^	nd	5.5 ± 0.0 ^a^	nd	5.0 ± 0.1 ^a^	24.3 ± 1.4 ^D^
*p*-coumaroyl aldaric acid	nd	nd	nd	nd	nd	6.7 ± 0.1 ^a^	6.7 ± 0.1 ^B^
Feruloyl methylaldaric acid	nd	nd	nd	20.0 ± 2.0 ^a^	nd	nd	20.0 ± 2.0 ^D^
Feruloylaldaric acid	nd	19.6 ± 1.6 ^c^	nd	1.2 ± 0.0 ^a^	nd	10.0 ± 0.8 ^b^	30.8 ± 1.7 ^E^
Feruloylaldaric acid	nd	9.7 ± 0.9 ^a^	nd	nd	nd	nd	9.7 ± 0.9 ^B^
Feruloyl methylaldaric acid	nd	8.1 ± 0.4 ^a^	nd	2.0 ± 0.2 ^a^	nd	1.5 ± 0.1 ^a^	11.6 ± 0.4 ^B^
Feruloyl methylaldaric acid	nd	20.4 ± 0.7 ^b^	nd	21.8 ± 2.9 ^a^	nd	10.2 ± 0.3 ^a^	52.4 ± 0.4 ^G^
Feruloyl methylaldaric acid	nd	15.8 ± 0.8 ^a^	nd	nd	nd	nd	15.8 ± 0.8 ^C^
1,3-coumaroyl-feruloyl-glycerol	nd	3.3 ± 0.1 ^a^	nd	nd	nd	nd	3.3 ± 0.1 ^A^
Total	42.5 ± 0.5 ^C^	90.7 ± 1.9 ^E^	nd	50.5 ± 1.2 ^D^	23.9 ± 2.0 ^A^	33.4 ± 0.4 ^B^	
Insoluble bound phenolics							
protocatechuic aldehyde	nd	28.6 ± 0.5 ^c^	nd	13.4 ± 0.4 ^a^	17.5 ± 0.3 ^b^	nd	59.4 ± 0.5 ^B^
*p*-hydroxybenzaldehyde	nd	nd	nd	nd	nd	nd	nd
2,4-dihydroxy benzoic acid	nd	17.2 ± 0.3 ^c^	nd	12.8 ± 0.8 ^b^	3.8 ± 0.1 ^a^	nd	33.8 ± 0.6 ^A^
*p*-coumaric acid	5.6 ± 0.4 ^a^	37.3 ± 0.2 ^c^	4208.1 ± 22.3 ^e^	50.7 ± 0.4 ^d^	24.3 ± 0.0 ^b^	nd	4326.1 ± 22.1 ^D^
ferulic acid	38.8 ± 1.3 ^b^	23.0 ± 0.1 ^a^	nd	40.0 ± 0.1 ^b^	39.8 ± 0.3 ^b^	39.5 ± 0.1 ^b^	181.2 ± 1.6 ^C^
Total	44.4 ± 1.7 ^B^	106.2 ± 1.0 ^D^	4208.1 ± 22.3 ^F^	116.9 ± 1.2 ^E^	85.5 ± 0.7 ^C^	39.5 ± 0.1 ^A^	

Key: protocatechuic aldehyde was quantified and expressed as protocatechuic acid equivalent; *p*-hydroxybenzaldehyde was quantified and expressed as *p*-hydroxybenzoic acid equivalent; coumaroylaldaric acids were quantified and expressed as *p*-coumaric acid equivalent; feruloyl aldaric acids, feruloyl methylaldaric acids, and 1,3-coumaroyl-feruloyl-glycerol were quantified and expressed as ferulic acid equivalent. Results were presented as means ± standard deviation of triplicates. Results were expressed as µg/g sample, dry weight basis. Values in a row with same superscript letters were not significantly (*p* < 0.0001) different from each other. nd = not detected. Significant differences among means of measured variables are shown by lowercase superscripts, while significant differences among means of total values are denoted by uppercase superscripts.

**Table 3 foods-10-02058-t003:** Flavonoid and 3-deoxyanthocyanidin composition of raw white rice, cowpea, sorghum leaves, their composite blend, and cooked products.

Sample	Raw Rice	Raw Cowpea	Raw Leaves	Raw Mix	Boiled Rice	Cooked Rice–Cowpea Blend	Total
Soluble free flavonoids and anthocyanins							
Catechin *O*-glucoside	nd	328.9 ± 14.9 ^c^	nd	168.7 ± 6.2 ^b^	nd	57.3 ± 1.1 ^a^	554.9 ± 7.7 ^C^
Procyanidin B1	nd	156.4 ± 0.9 ^c^	nd	70.2 ± 4.0 ^b^	nd	43.4 ± 0.9 ^a^	270.0 ± 4.1 ^B^
(Epi)-afzelechin *O*-glucoside	nd	255.9 ± 31.1 ^c^	nd	61.2 ± 5.5 ^a^	nd	105.9 ± 15.1 ^b^	422.9 ± 36.4 ^BC^
Quercetin dihexoside	nd	7.9 ± 0.2 ^c^	nd	3.8 ± 0.1 ^b^	nd	2.4 ± 0.2 ^a^	14.1 ± 0.4 ^A^
Quercetin hexoside	nd	16.4 ± 0.5 ^b^	nd	nd	nd	6.5 ± 0.3 ^a^	22.9 ± 0.5 ^A^
Naringenin	nd	nd	403.5 ± 4.1 ^b^	9.6 ± 0.2 ^a^	nd	nd	413.1 ± 4.1 ^BC^
Luteolinidin	nd	nd	3340.8 ± 308.0 ^b^	15.3 ± 0.1 ^a^	nd	nd	3356.1 ± 308.2 ^F^
Eriodictyol	nd	nd	2163.5 ± 171.9 ^c^	12.4 ± 0.3 ^b^	nd	5.4 ± 0.0 ^a^	2181.3 ± 171.6 ^E^
Apigeninidin	nd	nd	71,881.9 ± 351.6 ^c^	299.5 ± 2.7 ^b^	nd	3.4 ± 0.1 ^a^	72,184.8 ± 352.8 ^H^
Luteolin	nd	nd	1077.5 ± 36.2 ^b^	5.2 ± 0.2 ^a^	nd	1.5 ± 0.1 ^a^	1084.2 ± 36.1 ^D^
Apigenin	nd	nd	3630.3 ± 27.8 ^c^	17.7 ± 0.1 ^b^	nd	4.3 ± 0.4 ^a^	3652.3 ± 27.8 ^G^
Total	nd	765.5 ± 18.4 ^C^	82,497.6 ± 494.9 ^D^	663.6 ± 15.2 ^B^	nd	230.1 ± 5.2 ^A^	

Key: Results are presented as means ± standard deviation of triplicates. “nd” means not detected. Catechin *O*-glucoside and (epi)afzelechin *O*-glucoside were expressed as catechin equivalents. Quercetin dihexoside and quercetin hexoside were expressed as quercetin equivalents. Luteolinidin, apigeninidin, eriodictyol, naringenin, luteolin, and apigenin were expressed as naringin equivalents. Units of all measurements are expressed as µg/g sample, dry weight basis. Values in a row with different superscript letters were significantly (*p* < 0.0001) different from each other. Significant differences among means of measured variables are shown by lowercase superscripts, while significant differences among means of total values are denoted by uppercase superscripts.

**Table 4 foods-10-02058-t004:** Total phenolic content and antioxidant activities of raw white rice, cowpea, sorghum leaves, their composite blend, and cooked products.

Sample	RAW RICE	Raw Cowpea	Raw Leaves	Raw Mix	Boiled Rice	Cooked Rice–Cowpea Blend
Soluble free phenolics						
TPC	0.2 ± 0.0 ^a^	5.6 ± 0.2 ^d^	75.5 ± 1.2 ^e^	2.4 ± 0.0 ^c^	0.2 ± 0.0 ^a^	1.8 ± 0.1 ^b^
TEAC	1.8 ± 0.0 ^a^	51.7 ± 1.2 ^d^	1950.0 ± 208.4 ^e^	31.3 ± 0.5 ^c^	2.3 ± 0.0 ^a^	26.6 ± 1.3 ^b^
DPPH	0.6 ± 0.0 ^a^	20.6 ± 1.3 ^d^	595.4 ± 138.2 ^e^	10.5 ± 0.5 ^c^	0.5 ± 0.0 ^a^	7.5 ± 0.3 ^b^
ORAC	11.1 ± 0.7 ^a^	624.0 ± 112.0 ^d^	36,530.2 ± 761.6 ^e^	456.0 ± 38.0 ^c^	11.2 ± 0.2 ^e^	103.7 ± 1.3 ^b^
Total-free	13.7 ± 0.8 ^a^	701.9 ± 114.3 ^d^	39,151.1 ± 771.4 ^e^	500.2 ± 39.8 ^c^	14.2 ± 0.5 ^a^	139.6 ± 2.1 ^b^
Insoluble bound phenolics						
TPC	0.1 ± 0.0 ^a^	1.4 ± 0.3 ^c^	19.0 ± 1.0 ^d^	0.6 ± 0.1 ^b^	0.1 ± 0.0 ^a^	0.6 ± 0.0 ^b^
TEAC	1.9 ± 0.3 ^a^	18.0 ± 2.9 ^d^	307.7 ± 21.8 ^e^	6.5 ± 0.8 ^b^	1.6 ± 0.4 ^a^	9.8 ± 0.5 ^c^
DPPH	0.4 ± 0.0 ^a^	6.4 ± 0.9 ^c^	166.0 ± 16.2 ^d^	2.8 ± 0.7 ^b^	0.3 ± 0.0 ^a^	2.8 ± 0.1 ^b^
ORAC	80.6 ± 5.8 ^c^	180.7 ± 22.9 ^d^	3613.0 ± 76.8 ^e^	68.9 ± 5.7 ^b^	4.6 ± 0.3 ^a^	35.1 ± 0.7 ^b^
Total-bond	83 ± 5.8 ^c^	206.5 ± 23.1 ^d^	4105.7 ± 84.1 ^e^	78.8 ± 6.5 ^c^	6.6 ± 0.4 ^a^	48.3 ± 0.7 ^b^
Overall total	96.7 ± 5.9 ^b^	908.4 ± 115.8 ^e^	43,256.8 ± 780.6 ^f^	579.0 ± 40.4 ^d^	20.8 ± 0.6 ^a^	187.9 ± 2.4 ^c^

Key: Results are presented as means ± standard deviation of triplicates. TPC—total phenolic content, expressed as mg FAE/g sample, dry weight basis; DPPH—2,2-diphenyl-1-picrylhydrazyl, expressed as µmol TE/g sample, dry weight basis; TEAC—Trolox equivalent antioxidant capacity, expressed as µmol TE/g sample, dry weight basis; ORAC—oxygen radical absorbance capacity, expressed as µmol TE/g sample, dry weight basis. FAE—ferulic acid equivalent; TE—Trolox equivalent. Values in a row with different superscript letters were significantly (*p* < 0.0001) different from each other.

**Table 5 foods-10-02058-t005:** Percentage contribution of each sample to the principal components for the principal component analysis.

Sample	Raw Rice	Raw Cowpea	Raw Leaves	Raw Mix	Boiled Rice	Cooked Rice–Cowpea Blend
Principal component 1	2.21788	6.39497	82.70456	3.72584	2.27364	2.68310
Principal component 2	19.89433	63.53699	0.62503	1.11919	9.99322	4.83123
Principal component 3	18.99977	3.14427	0.00002	2.28827	11.44832	64.11934
Principal component 4	20.59325	8.62702	0.00091	49.92923	11.83996	9.00963
Principal component 5	21.62809	1.63008	0.00280	26.27080	47.77819	2.69004

## Data Availability

Not applicable.

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
