# Peer review of "Phenolic Composition and Antioxidant Properties of Cooked Rice Dyed with Sorghum-Leaf Bio-Colorants"

_foods, 2021, doi:10.3390/foods10092058_

Round 1

Reviewer 1 Report

The manuscript titled "Phenolic composition and antioxidant properties of cooked rice dyed with sorghum-leaf bio-colorants" by Franklin Brian Apea-Bah , Xiang Li , and Trust Beta is well-composed and prepared in a good flow. However, there are issues (see below) that need authors' careful considerations.

  • Section 2.4: All the LC-conditions applied during separations including column type, mobile phase, gradient, flow, temperature, and volume should be specified along with key mass parameters. 
  • Along with figures 1-4, it would be great if the authors add a chromatogram for a mixture of authentic standards. Together, the baseline of the chromatogram in Figure 1A should be made visible. Looking at the chromatograms in these figures, however, it seems that the LC-condition(s) is/are poorly optimized. What is the authors' take on this issue? 
  • Line 87-89, 120-121: The authors stated that they identified the compounds using retention times of standards, UV, and mass spectral data along with literature. This is a mere description and hence, a more elaborated analysis related to the structural identification process of each molecule should be provided (this can be part of the main body or provided as supplementary material). The literature used as references should also be cited in Table 1 where appropriate.  
  • Line 60: The phrase "all other chemicals" should be removed and replaced by the list of all the chemicals.
  • While results under section 3.1 are relatively well composed and analyzed, results under section 3.2 are poorly analyzed and compared with previous studies (the authors are short of literature between lines 255-281). This needs a thorough revision.
  • The list/names of compounds should be consistent/uniform throughout the manuscript including tables 1, 2, and 3, as well as figure footnotes. 
  • Table 2 (Insoluble bound phenolics section): The assignment of superscript letters for the level of significance should be revised (mainly for compounds p-coumaric acid and ferulic acid, and total content).
  • Tables 2 and 3 should specify "nd" as part of footnotes.
  • The legend of Figure 6 says "Bi-plot for the factor coordinates....". There is also a similar description in line 287. But, the figure shows separate loading and score plots and hence, the legend should be revised or a PCA bi-plot should be provided. Besides, the score plot is cut and not fully visible and therefore, should be reloaded.
  • To meet the objective of the research, the PCA should be computed or described again in a way that shows the contribution of variables (phenolic compounds) to characterize/separate the samples, not the other way round (Table 5, Line 292-293; 309-310). 
  • ml should be changed to mL throughout the manuscript.
  • Is the conclusion Section 5? If so, section 4 is missing in the manuscript. The last sentence in the conclusion part (line 343)  seems unfinished and hence, needs revision. 
  • The page number is missing in reference 1 (line 362).

Author Response

Reviewer comment is attached

Reviewer 2 Report

The presented manuscript is well written and contains valuable results of research on the possibility of increasing the content of phenolic compounds and antioxidant activity in cooked rice by cooking it with red cowpea seeds and dye sorghum leaves. I have only a few comments and quastions.

Did the authors check how cooking of rice with cowpea seeds and dye sorghum leaves influence on sensory attributes of rice and consumers acceptance?

Line 109: Is it a sense to establish p-value at 0,0001?

Figure 6 is unreadable.

In conclusion “p-cumaric” “p” – should be written in Italic.

Table 2, last line: It should be β or B.

Author Response

Reviewer comment is attached

Round 2

Reviewer 1 Report

Authors, you have addressed most of the issues raised during the first round of revision, and congratulations on that. However, the following very important points must still be clarified. 

  • As stated between lines 147-152, compounds with no available authentic standards are identified using literature. Though you are not able to provide details of the identification process,  you still need to explain how you identified the isomeric compounds (Table 2). For instance compounds 8 & 9, compounds 10, 11, & 12, compounds 13-18, etc.  Otherwise,  it is difficult to say that 27 different phenolic compounds are identified and quantified in your study. Again, the mass data provided in Table 1 (i.e the identification process) should be supported by literature/references. 
  • How are those compounds (with unavailable standards) quantified? Also, the number of standards used (Figures 7 and 8) does not match with the number of compounds quantified (Tables 2 and 3). Therefore,  taking these mismatches in to considerations, the quantification description given between lines 148-150 should be well clarified. If representative standards are used to quantify a group of compounds, as you tried to show in the figure legends (figures 7 and 8) and table footnotes (table 3), that also should be elaborated in the manuscript. For instance,  which standard is used to quantify eriodictyol? apigeninidin? etc.
  • Figure 2 (and Table 3): Among compounds 11 and 12, one must be a hexoside derivative (not dihexoside) since it doesn't support the mass data in table 1. Similarly, in figures 3 and 4, either compound 10 or 13 should be apigenin (both are identified as apigeninidin). Again, compound no. 3's name in table 1 should match those in figures 2, 3, and 4.  Such misrepresentations of compounds should be rechecked throughout the manuscript. 
  • Figure 3 and 4: peak 7 (unknown) should be removed. It is niether quantified nor its mass data is provided in Table 1. What makes it different from the rest unidentified peaks in those figures?
